# Role of Epithelial-to-Mesenchymal Transition for the Generation of Circulating Tumors Cells and Cancer Cell Dissemination

**DOI:** 10.3390/cancers14225483

**Published:** 2022-11-08

**Authors:** Gaetan Aime Noubissi Nzeteu, Claudia Geismann, Alexander Arlt, Frederik J. H. Hoogwater, Maarten W. Nijkamp, N. Helge Meyer, Maximilian Bockhorn

**Affiliations:** 1University Hospital of General and Visceral Surgery, Department of Human Medicine, University of Oldenburg and Klinikum Oldenburg, 26129 Oldenburg, Germany; 2Laboratory of Molecular Gastroenterology & Hepatology, Department of Internal Medicine I, UKSH-Campus Kiel, 24118 Kiel, Germany; 3Department for Gastroenterology and Hepatology, University Hospital Oldenburg, Klinikum Oldenburg AöR, European Medical School (EMS), 26133 Oldenburg, Germany; 4Section of HPB Surgery & Liver Transplantation, Department of Surgery, University Medical Center Groningen, University of Groningen, 9713 GZ Groningen, The Netherlands

**Keywords:** epithelial-to-mesenchymal transition (EMT), circulating tumors cells, metastasis, metastatic cascade, EMT transcription factors

## Abstract

**Simple Summary:**

It is well established that CTCs play an important role in tumor diagnostics and prognosis; this review outlines the mechanisms responsible for their generation, their roles in cancer dissemination and metastasis, and their clinical applications in precision medicine.

**Abstract:**

Tumor-related death is primarily caused by metastasis; consequently, understanding, preventing, and treating metastasis is essential to improving clinical outcomes. Metastasis is mainly governed by the dissemination of tumor cells in the systemic circulation: so-called circulating tumor cells (CTCs). CTCs typically arise from epithelial tumor cells that undergo epithelial-to-mesenchymal transition (EMT), resulting in the loss of cell–cell adhesions and polarity, and the reorganization of the cytoskeleton. Various oncogenic factors can induce EMT, among them the transforming growth factor (TGF)-β, as well as Wnt and Notch signaling pathways. This entails the activation of numerous transcription factors, including ZEB, TWIST, and Snail proteins, acting as transcriptional repressors of epithelial markers, such as E-cadherin and inducers of mesenchymal markers such as vimentin. These genetic and phenotypic changes ultimately facilitate cancer cell migration. However, to successfully form distant metastases, CTCs must primarily withstand the hostile environment of circulation. This includes adaption to shear stress, avoiding being trapped by coagulation and surviving attacks of the immune system. Several applications of CTCs, from cancer diagnosis and screening to monitoring and even guided therapy, seek their way into clinical practice. This review describes the process leading to tumor metastasis, from the generation of CTCs in primary tumors to their dissemination into distant organs, as well as the importance of subtyping CTCs to improve personalized and targeted cancer therapy.

## 1. Introduction

Metastasis, i.e., the spread of cancer cells from their site of origin to distant tissues or organs, is the leading cause of death in almost all solid cancers [1]. However, the cellular and molecular mechanisms that promote tumor cell metastasis are not fully understood. The formation of metastases is mainly driven by the dissemination of cancer cells that are shed from the primary tumor intravasate and circulate in the bloodstream—so-called circulating tumor cells (CTCs). Consequently, CTCs have to resist blood pressure and defend against patrolling immune cells. Thus, extravasation tumor cells have to adapt to the cellular environment of the tissue or organ of the colonization site. Transformation into CTCs requires a number of biochemical and cellular changes that facilitate migration, invasion, resistance to apoptosis, and the modulation of extracellular matrix (ECM) production [2]. During epithelial-to-mesenchymal transition (EMT), ECM is reorganized in such a way that epithelial cells lose cell–cell as well as cell–matrix contacts. Cell adhesion molecules, particularly E-cadherin, which significantly contributes to the epithelial cell junction and apico-basal polarization [3,4,5], are downregulated. Conversely, vimentin, fibronectin, and N-cadherin are upregulated, fostering the transition into a motility state. Overall, cells lose epithelial characteristics while progressively adopting a quasi-mesenchymal phenotype. This cellular program typically is fully reversible and mesenchymal-to-epithelial transition (MET) allows CTCs to regain their initial epithelial phenotype that is required for metastasis formation [5]. In conclusion, EMT and MET—originally implicated in embryogenesis and adult wound healing—play crucial roles in metastatic dissemination by CTCs [6]. Furthermore, EMT is critical for the acquisition of cancer stem cell (CSC)-like properties including self-renewal, resistance to apoptosis, tumor progression, and metastasis. This may result in the generation of cancer cell progenitors with strong drug resistance, e.g., via the expression of membrane protein transporters such as the ATP-binding cassette [7,8].

This review discusses current advances in the understanding of the EMT/MET of CTCs with regard to cancer diagnosis, prognosis, and therapy as well as their potential application in precision medicine

## 2. Molecular Basis of Epithelial-to-Mesenchymal Transition in the Generation of Circulating Tumors Cells

The presence of CTCs in the bloodstream, either as individual cells or cell clusters, is regarded as an early sign of cancer progression. Particularly, it was found that clusters of CTCs display elevated hypomethylation on transcription factor binding sites, involved in stemness, proliferation, and cell adhesion, all of which were also implicated in EMT [9]. EMT is an evolutionary conserved developmental program involved in gastrulation, mesodermal development, and neural crest formation. Furthermore, it is essential for tissue regeneration, fibrosis, and wound healing. In solid cancers, EMT is critical for the generation of CTCs and the progression and metastasis of tumors [10,11]. It is tightly regulated by multiple signaling pathways, including stem cell factor (SCF), platelet-derived growth factor (PDGF), vascular endothelial-derived growth factor (VEGF), epidermal growth factor (EGF), hepatocyte growth factor (HGF), fibroblast growth factors (FGF), bone morphogenetic proteins (BMPs), transforming growth factor (TGF)-β, hypoxia-inducible factor 1-α (HIF1α), Wnt, and Notch signaling. Particularly, cancer-associated fibroblasts (CAFs) that predominantly accumulate in the stroma of the tumor microenvironment (TME) of gastric (GC), pancreatic (PC), and hepatic cancers stimulate the EMT of cancer cells via interleukin-6 and TGF-β1 secretion. Through the release of cytokines and growth factors, CAFs also significantly contribute to ECM synthesis and remodeling, which play a key role in tumor cell dissemination. In addition, CAFs stimulate angiogenesis and tumor cell proliferation [12,13,14]. The generation of CTCs is then governed by several transcription factors which directly or indirectly regulate EMT, namely, the twist basic helix–loop–helix transcription factor 1 (TWIST1), the Snail homologs Snail 1 and 2, and zinc finger E-box binding homeobox (ZEB) 1 and 2 [3,15]. These EMT-associated transcription factors (EMT-TFs), also regarded as master regulators of EMT, promote the repression of the epithelial markers E-cadherin, RKIP, mucin-1, claudin, occludin, and PTEN, while inducing the expression of the mesenchymal markers N-cadherin, vitronectin, vimentin, and various matrix metalloproteases (MMPs) [8,16].

## 3. Role of Circulating Tumors Cells with Cancer Stem Cell Properties in Metastasis

Similar to cells of the primary tumor, CTCs are extremely heterogeneous and contain phenotypically and genetically distinct subpopulations including epithelial cells (E-CTCs), epithelial-to-mesenchymal transition cells (EMT-CTCs), hybrid epithelial/mesenchymal cells (EM-CTCs), mesenchymal cells (M-CTCs) and CTCs with stem-cell like properties, termed circulating tumor stem-like cells (CTSCs) [17]. The majority of CTCs, however, cannot survive in the bloodstream and are unable to seed metastases, suggesting that metastatic colonization is a remarkably inefficient process [18]. CTSCs are irregular cells whose phenotype may vary between cancer subtypes and even within the same type of cancer [19]. Owing to their CSC-like properties, CTSCs can escape anoikis [20], are more likely to survive in the bloodstream, and spread metastatically. As they also tend to be more resistant to conventional therapeutic strategies, they are associated with cancer relapse [21]. It is now widely accepted that CTSCs are involved in tumor development, metastasis, and the therapy resistance of solid cancers, including GC, BC, CRC, hepatocellular carcinoma (HCC) and PC. Furthermore, CTSCs and CTCs have been used to diagnose and predict patient outcomes [22,23,24].

Metastasis requires tumor cells to go through a series of events, namely, invasion, intravasation, surviving bloodstream circulation, extravasation, and colonization, commonly referred to as metastatic cascade (Figure 1).

### 3.1. Metastatic Cascade

#### 3.1.1. Invasion and Intravasation

The metastatic cascade starts with the invasion of the tumor into the surrounding tissue. Via EMT, epithelial tumor cells can acquire a more invasive, stem-cell-like phenotype allowing them to detach from the primary tumor. The elevated release of MMPs leads to a modulation of ECM and further facilitates invasion [19]. The production of proangiogenic factors such as VEGF promotes angiogenesis. Additionally, CSCs can transdifferentiate into endothelial cells to support the formation of blood vessels [25]. It was shown that endothelial cells in glioblastoma, a highly angiogenic malignancy, shared approximately 60% of their genomic alterations with tumor cells, suggesting that the vascular endothelium originated from the neoplasia. Indeed, it was found that a subcutaneous injection of freshly isolated CD133^+^/CD31^−^ glioblastoma stem-like cells resulted in human CD31 expression, an endothelial marker, in xenograft tumor tissues of mice [26]. The transdifferentiation of glioblastoma stem-like cells into endothelial-like cells was also confirmed in vitro [27].

Pericytes are multi-functional mural cells of the microcirculation and overexpress the transmembrane receptor endosialin. Thereby, these cells promote cell–cell contact with tumor cells and foster their migration through the endothelium. In the TME, pericytes are also the main source of IL33, and thus responsible for recruiting tumor-associated macrophages (TAMs) via the IL33-ST2 signaling pathways [28,29]. Moreover, monocytes recruited by CCR2 signaling can differentiate into TAMs. CXCL12 released by perivascular fibroblasts facilitates the co-migration of tumor cells and TAMs into blood vessels. The differentiation of TAMs into perivascular macrophages stimulates vascular leakage, further promoting intravasation [30].

#### 3.1.2. Surviving Bloodstream Circulation

After intravasation, CTCs face a hostile environment that poses both physical and biological threats [31]. Above all, circulating in the bloodstream subjects cells to shear stress. As CTCs are poorly adapted to this kind of mechanical force [32], most of them will be directly destroyed or enter apoptosis. Additionally, CTCs have to withstand the constant attacks of immune cells. The interplay of internal and external factors critically determines whether a cell can survive in the bloodstream. Internal factors include genetic alterations and the abnormal gene expression of, e.g., immunomodulatory factors and apoptosis inhibitory factors such as survivin or stem cell-like characteristics. External factors involve immune cells, cytokines, platelets, and circulating tumor microemboli that may protect CTCs from cellular stress. Among others, tissue hypoxia and autophagy may also impact cell survival [33]. CTCs can undermine immunological clearance through several different strategies. They often overexpress immunosuppressive proteins such as the immune checkpoint PD-L1, e.g., in BC, esophageal cancer, CRC, and PC [34,35]. The immune checkpoint V-domain Ig suppressor of T-cell activation (VISTA) is upregulated in lymphocytes from metastatic melanoma patients, suggesting that VISTA might be implicated in metastasis [36]. In addition, VISTA was highly expressed in advanced stages of ovarian cancer and particularly dominant in lymph node metastases. Recently, it has been shown that VISTA inhibits T-cell effector functions and the release of IL2 from CD4+ T cells by suppressing granzyme B release from cytotoxic T cells [37,38]. Moreover, platelets seem to have a significant role in the immune evasion of CTCs, as these cells attract platelets by the expression of coagulation factors, for instance, thromboplastin. Being fully covered with platelets protects CTCs from NK and other immune cells and reduces the shear stress on CTCs [39,40]. Platelet-derived TGF-β and direct platelet–tumor cell contact activates the TGF-β/Smad and NF-κB pathways in a synergistic manner, which promotes the acquisition of a mesenchymal-like phenotype [41]. Furthermore, CD47 expressed on the surface of CTCs inhibits the interaction with macrophage-expressed signal-regulating protein alpha (SIRPα) and subverts the phagocytic machinery [42,43]

#### 3.1.3. Extravasation and Colonization

The small number of CTCs that are able to withstand blood circulation can eventually settle in a distant organ after extravasation [44]. In 1889, Paget shaped the seed and soil hypothesis, according to which CTCs and the host organ must have a certain degree of compatibility to allow for the formation of metastases [45]. Albeit challenged over the last century, Paget’s theory seems to be supported by recent research on the mechanisms of metastatic diseases. However, the initial theory has been extended by the concept of pre-metastatic niche formation: prior to the dissemination of tumor cells, the primary tumor can induce the formation of a supportive and receptive tissue microenvironment via the production of tumor-derived secreted factors VEGF-A, tumor necrosis factor α (TNF-α) and TGF-β. As a result of this preparation, tumor cells settle more effectively owing to an accumulation of compromised immune cells and ECM [46]. Some solid cancers have preferred organs for metastasis, e.g., PC prefers to metastasize to the liver and peritoneum, with more than 40% preference for those organs and about 13% preference for the lung. CRC and GC are even more inclined to metastasize to the liver with about 70% and 48%, respectively, making the liver the most common organ for those solid cancers to metastasize [47,48,49]. Prior to extravasation, CTCs adhere to the endothelium via CD44, integrin β1, integrin αvβ3, and α5β1 [19,50]. CTCs release glycocalyx that enhances integrin binding in order to promote intravascular adhesion and extravasation [51]. It has been demonstrated that metastatic colonies originate from intravascular tumor cells [52], which in the last step of the metastatic cascade, the extravasation, migrate through the endothelium of the blood vessel and colonize the pre-metastatic niche [44]. In addition, exosomes, which facilitate the communication of primary tumor cells with the microenvironment of distant organs, are implicated in pre-metastatic niche formation [53]. A role of exosomes as a key factor for the (epigenetic) reprogramming of cells in targeted organs has been reviewed elsewhere [54]. It seems likely that CTCs need to acquire stem cell-like properties for successful colonization. Recently, cellular and transcriptomic changes were observed during in vivo metastasis formation in a clonal outgrowth model of patient-derived CRC organoids in mice. Peculiarly, micrometastatic lesions were devoid of CSCs, but on the other hand, de novo CSCs were indeed present in overt metastases [55]. After extravasation, metastatic colonization is further stimulated by MET, which triggers the re-expression of E-cadherin and the downregulation of Prrx1, which subsequently decreases Snail 1 expression and leads to the re-acquisition of an epithelial phenotype and epithelial morphology [56,57].

## 4. Clinical Applications of CTC Enumeration and CTC Subtyping

CTCs have been extensively studied as clinical biomarkers in cancer diagnostics and suggested as a predictive marker, including the prediction of metastasis and the prognosis of PC, GC, CRC, BC, and esophageal cancer [58,59]. The presence of CTCs before surgery was significantly related to poor overall and shorter recurrence-free survival, as well as higher disease recurrence rates in patients with non-metastatic CRC [60]. CTC counts at baseline and during therapy for advanced gastric and esogastric cancer were useful in predicting the therapeutic effectiveness. The presence of CTCs in patients with esophageal cancer was closely linked to shorter overall and relapse-free survival [59,61]. In PC, it was observed that patients with CTCs detected in the portal vein were more likely to develop liver metastases after surgery than patients with CTCs detected in the peripheral blood. CTCs also predict early distant metastasis and impaired survival in resected PC [62,63]. In patients with HCC, microvascular invasion is significantly associated with CTC numbers in the systemic circulation [64]. Overall, in addition to being used in cancer diagnosis and prognosis, CTCs can also be used to monitor therapeutic response and to predict therapy success [65].

However, CTCs are difficult to detect and analyze, particularly due to their genetic and phenotypic heterogeneity and their scarcity in blood samples. Whole blood can contain less than 100 and sometimes even less than 10 clusters of CTCs per 10^7^ leukocytes and 500 × 10^7^ erythrocytes [66,67]. Despite being a rare event, CTCs offer great potential in liquid biopsy, which allows safe, inexpensive, and reproducible sampling as an alternative to invasive biopsy [68]. In clinical applications, CellSearch (Menarini Silicon Biosystems, Bologna, Italy) is the most commonly used platform for the isolation and enrichment of CTCs. In 2004, CellSearch was cleared by the FDA for the detection and enumeration of CTCs as a predictor of progression-free survival in BC. Furthermore, it was approved to detect prostate cancer, CRC, and BC [69], and has shown clinical relevance for many other cancers including HCC, PC, and GC [62,70,71,72].

CellSearch is based on the immunomagnetic enrichment of EpCAM^+^ CTCs from blood samples. CTC identification relies on cell morphology and the immunofluorescence staining of cytokeratin (CK), whereas CD45, commonly expressed on blood leukocytes, is used as a negative marker [73]. According to the manufacturer’s protocol, a CTC is defined as CK+ and CD45−, has a visible nucleus, and is morphologically round to oval [74]. To date, a variety of different CTC detection platforms are commercially available, many of which still rely on EpCAM-dependent CTC enrichment (Table 1). Historically, epithelial cell markers have been utilized for CTC detection, as epithelial cells are typically absent in the bloodstream of a healthy individual [72]. However, this strategy comes with a severe drawback: lack of or reduced EpCAM expression has been found in the CTCs of various cancers including PC, and the number of E-CTCs does not always correlate with clinical parameters [75]. In patient-derived xenograft models for breast, liver, prostate, and lung cancer, variable EpCAM expression has been detected [76]. Moreover, a study confirmed the presence of EpCAM^-^ and EpCAM^low^ CTCs in metastatic LC that were disregarded by the CellSearch platform. Considering the CTCs found in the CellSearch waste, the percentage of patient samples with ≥1 CTC in 7.5 mL of blood increased from 41% to 74% [77]. Consequently, more and more techniques are developed that allow the marker-independent isolation of CTCs and are commercially available (Table 1). These isolation strategies opened up a whole new avenue for novel CTC detection markers.

As EMT plays a major role in the intravasation of CTCs and their metastatic capability in circulation, the analysis of CTC subtypes bares great potential to reveal the molecular mechanisms underlying metastasis formation and promote the development of novel diagnostic and therapeutic approaches. The analysis of surface markers on subpopulations of tumor cells that undergo spontaneous EMT in a genetic mouse model of skin squamous cell carcinoma revealed different EMT transition states, including hybrid states, corresponding to tumor cells exhibiting both epithelial and mesenchymal characteristics [100]. It is now well established that EMT occurs through distinct intermediates, referred to as E/M or partial EMT states, in vivo [101,102]. In a recent study, the clinical relevance of three different CTC subtypes (Figure 2)—E-CTCs, M-CTCs, and E/M-CTCs—were evaluated in PC patients using CanPatrol [103]. Particularly, the presence of M-CTCs was significantly associated with aggressive disease, distant metastasis, and tumor node metastasis (TNM) stage [103]. In a similar study, CTCs from the blood of PC patients were analyzed and phenotypically categorized using dielectrophoresis flow fractionation in combination with Imagestream flow cytometry. In addition to E-CTCs, E/M-CTCs—referred to as partial (p)EMT-CTCs—and M-CTCs subtypes, a CTSC subtype was also identified. Peculiarly, the most prevalent subpopulation in PC was pEMT-CTCs, which were associated with overt metastases and early recurrence after resection [104].

Considering the EMT-dependent formation of CTCs in PC as well as other solid cancers, therapeutically targeting EMT in the TME seems well justified. EMT has been successfully targeted *in vitro*, either directly or indirectly. Blocking the PI3K/Akt signaling pathway has been shown to suppress EMT in an HCC cell line via the Snail/GSK-3/beta-catenin pathway [105]. Forkhead box protein C2 (FOXC2) is an essential factor for the initiation and maintenance of EMT. Consequently, MC-1-F2, a FOXC2 inhibitor, induced cadherin switching and reverse EMT, decreasing the migration and invasion of a BC cell line [106]. Strategies to therapeutically target EMT in cancer have been comprehensively discussed elsewhere [107,108,109]. Intriguingly, various druggable EMT targets are upregulated on CTCs. The receptor-like tyrosine kinase-like orphan receptor 1 (ROR1) is an embryonic protein that plays an important role in cell proliferation, differentiation, and angiogenesis, but is also overexpressed in cancer and induces tumor migration and metastasis. In BC cell lines, ROR1 was successfully targeted to attenuate EMT. Additionally, BC adenocarcinomas expressing high levels of ROR1 had a higher chance to display gene expression signatures related to EMT [110]. Interestingly, ROR1 was elevated in CTCs of PC patients, promoting their proliferative and invasive capacity [111]. Promising results have also been obtained targeting the mesenchymal–epithelial transition factor c-Met, a receptor tyrosine kinase that is often overexpressed in metastatic non-small cell lung cancer (NSCLS). Consequently, the c-Met inhibitor savolitinib has been conditionally approved in China for the treatment of metastatic NSCLC. Additionally, a phase II study is also being conducted on savolitinib to treat GC and papillary renal cell carcinoma (NCT04923932) [112]. Interestingly, c-Met has also been suggested as a marker for CTC isolation [113]. Furthermore, the metastatic spread has been found to be associated with an upregulation of the mitotic and apoptotic transcription factor survivin (BIRC5) on CTCs. Consequently, pharmacologic inhibition of survivin with YM155 or survivin knockdown triggered cell death and anoikis in tumor organoid models. Thus, survivin may facilitate the survival of CTCs in the circulation. [114,115]. Overall, these observations suggest that beyond CTC enumeration as a mere diagnostic, prognostic, or predictive tool, targeting CTCs bares an unused therapeutical potential and could thereby substantially increase the effectivity of cancer therapy.

## 5. Conclusions

Metastasis and tumor formation are largely influenced by EMT. During EMT, epithelial cells are transformed into mesenchymal cells that can also acquire stem cell-like properties, have increased motility and invasiveness, and display resistance to multiple therapeutic strategies. CTCs generated by EMT represent a reservoir source of information essential to cancer diagnostic, therapeutic, and precision medicine. However, in many cancer patients, CTC counts in the peripheral blood range from only 1 to 10 cells per 10 mL, which makes an analysis of these cells extremely challenging. Since most of the CTC detection systems are based on EpCAM, a marker on the epithelial cell surface that is decreased during EMT, new methods are being explored to detect and enrich CTCs. Furthermore, heterogeneity is another challenge for liquid biopsy. Forthcoming studies should carefully adjust clinical criteria to account for CTC heterogeneity.

Overall, it is possible to decrease cancer mortality by developing new technology to identify the most aggressive subsets of CTCs and by optimizing the analytical sensitivity of the method of choice. Early elimination of these CTCs offers a novel strategy to overcome metastatic disease.

## Figures and Tables

**Figure 1 cancers-14-05483-f001:**
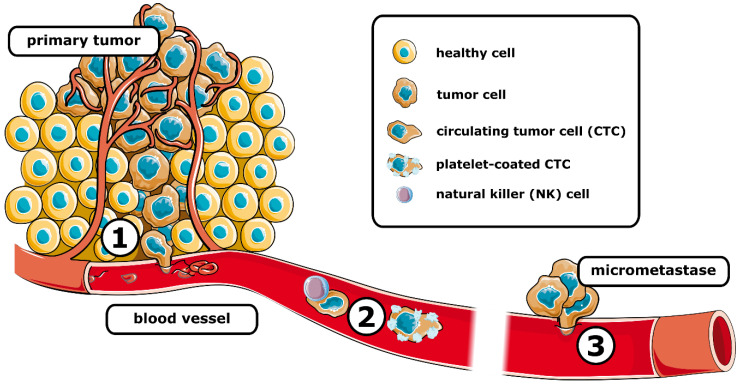
Schematic representation of the metastatic cascade. After intravasation (1) tumor cells travel the systemic circulation (2), where they are susceptible to elimination, e.g., by natural killer cells. Coating by platelets can protect circulating tumor cells. After extravasation, the (3) formation of micrometastases is associated with the colonization of distant tissue. (See text for details).

**Figure 2 cancers-14-05483-f002:**
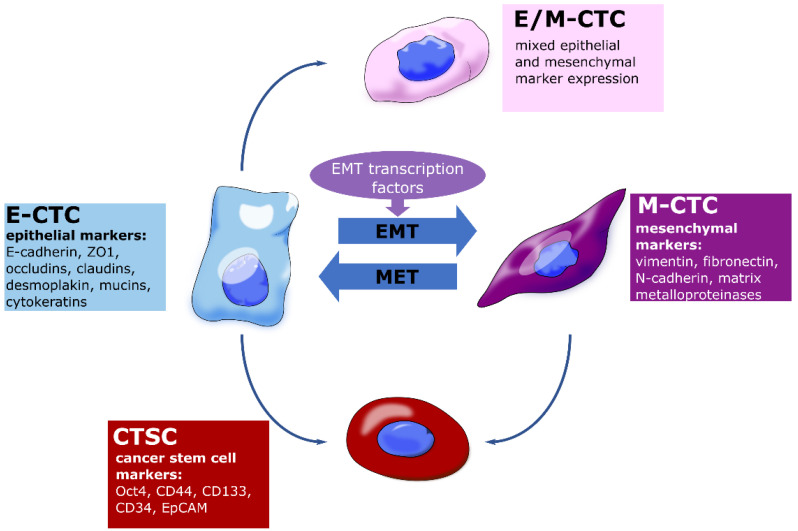
Clinically relevant CTC subtypes. (See text for details).

**Table 1 cancers-14-05483-t001:** Common methods for isolation and analysis of CTCs.

Name	Type of Method	Target	Cancer	Clinical Trials	Reference
CellSearch^®^ system	Positive IMS	EpCAM	GC, BC, PC	NCT01116869	[63,74,78]
EPHESIA CTC-Chip	Positive IMS	EpCAM	PCa, BC		[79]
MagSweeper™	Positive IMS	EpCAM	BC		[80,81]
Herringbone-Chip	Positive IMS	EpCAM	PC, LC, NSCLC	NCT01734915	[82]
IsoFlux	Positive IMS	EpCAM	PC, CRC, PCa		[83]
Adnatest^®^ system	Positive IMS	EpCAM, MUC-1 HER2, CEA	BC, CRC		[84]
Nanovelcro-chip	Positive IMS	EpCAM	PCa		
GEDI microdevice	Positive IMS	PSMA	PCa		[85]
CTC-iChip	Negative IMS	CD45, CD66b	BC		[86,87]
ClearCell^®^ FX	Dean flow fractionation	Label-free	BC, PC, OC	NCT04696744	[88,89,90]
ApoStream^®^	Dielectrophoretic field -flow fractionation	Electrical signature	BC, LC, PC	NCT02349867	[91,92]
Onco-Quick^®^	Porous membrane	Density, size	BC, Carcinoma		[93,94]
VITA-Assay^TM^	Positive IMS	Cell adhesionmatrix (CAM)	PCa		[95]
ISET^®^	Microfilter	Size, shape	LC, BPC, MPM	NCT03328559, NCT01776385	[96]
ScreenCell	Microfilter	Size, shape	BC, EA	NCT02610764	[97]
EPIDROP	Positive IMS	Variable	PC, PCa	NCT05346536, NCT04581109 NCT04556916	[98] (review)
EPISPOT	Protein secretion epithelial immunoSPOT	CK19, MUC-1, PSA	BC, CRC, PCa	NCT01596790, NCT01402154	[98,99]

Abbreviations: gastric cancer (GC), breast cancer (BC), pancreatic cancer (PC), prostate cancer (PCa), lung cancer (LC), non-small cell lung cancer (NSCLC), oropharyngeal cancer (OC), bronchial primitive cancer (BPC). Malignant pleural mesothelioma (MPM), esophageal adenocarcinoma (EA), immunomagnetic separation (IMS).

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
