# Peer review of "Role of Epithelial-to-Mesenchymal Transition for the Generation of Circulating Tumors Cells and Cancer Cell Dissemination"

_cancers, 2022, doi:10.3390/cancers14225483_

Round 1

Reviewer 1 Report

In my opinion the review covers a topic too general as EMT and metastasis involve multiple stages each of them with their molecular, cellular and therapeutical features. As a result, the review does not seem to deal with new knowledge, especially in the description of the EMT process from the molecular point of view. Many of the references are a bit old and/or are, in turn, reviews.

Minor points:

In section 2, the EMT and CTCs parts should be better linked.

In point 3.1.3. the author should talked about the role of exosomes in preparation of the metastatic niches as this could add novelty to the review.

In section 4. the therapeutical approaches targeting EMT seem too short and skewed and deserve a more extended discussion.      

Reviewer 2 Report

In this review by Aime Noubissi Nzeteu et al entitled “Role of Epithelial to mesenchymal Transition for the Generation of Circulating Tumors Cells and Cancer Cell Dissemination”, the authors provide a brief review on some of the numerous events that take place during solid cancer metastasis. The main piece of work is broken down into sections discussing the process of EMT and primarily the different growth factors regulating the process, as well as cancer stem cells and the process of metastasis.

The information provided is relevant to cancer and offer some background about the some aspects of cell dissemination.

However EMT and transcription factors regulating the carcinogenesis and metastasis have been shown to be key regulators of cancer progression for many years, if not decades, and this review only scratches the surface on this topic, making it somewhat simplistic compared to much broader analysis and reviews recently published. The same exact comments can be made about cancer stem cells or the section related to the metastasis process.

Given that a large body of information is used to discuss circulating tumour cells and their subtyping in the latter part of the manuscript, perhaps this review should focus on this element of the work and ignore the other 2-3 elements (EMT, metastasis, stem cell etc…) which form 50% of the work and in doing so distract the readers.

Reviewer 3 Report

The author, in this review, describes to the importance of biological and pathological function of EMT in the tumor progression. The paper is well organized and well written. Some minor comments are listed below.

1. Line 77, TGF-b should be spelled out at the first appearance in the main text.

2. Line 359, partial (p)-EMT is denoted, but its details are missing. In addition to partial EMT, hybrid phenotypes are also recently associated with CTC and CSC (Nature. 2018 Apr;556(7702):463-468. doi: 10.1038/s41586-018-0040-3. Trends Cell Biol. 2019 Mar;29(3):212-226. doi: 10.1016/j.tcb.2018.12.001. Nat Rev Mol Cell Biol. 2020 Jun;21(6):341-352. doi: 10.1038/s41580-020-0237-9.). 

3. Concerning platelet-coated CTC in figure 1, briefly mention it in the main text and cite the following reference.

Cancer Cell. 2011 Nov 15;20(5):576-90.  doi: 10.1016/j.ccr.2011.09.009.

Reviewer 4 Report

This was an informative and enjoyable review to read.  It nicely summarizes circulating tumor cell development and detection.

Author Response

We thank the reviewer for the kind remarks.

Round 2

Reviewer 1 Report

No comments

Reviewer 2 Report

Changes made by the authors are not appropriate and the focus of the work reviewed is suitable for publication